# Study of Bayesian variable selection method on mixed linear regression models

Yong Li[1]*, Hefei Liu[1], Rubing Li[2]

**1** School of Mathematics and Statistics, Qujing Normal University, Qujing, China, **2** School of Economics, Shanghai University of Finance and Economics, Shanghai China

\* qjsfxyly@163.com

**Data Availability Statement:** All relevant data are within the paper and its Supporting information files.

**Funding:** This article is partially supported by the fund of Yunnan Provincial Science and Technology Department of China (Award Number: 2019FH001-

## Abstract

Variable selection has always been an important issue in statistics. When a linear regression model is used to fit data, selecting appropriate explanatory variables that strongly impact the response variables has a significant effect on the model prediction accuracy and interpretation effect. redThis study introduces the Bayesian adaptive group Lasso method to solve the variable selection problem under a mixed linear regression model with a hidden state and explanatory variables with a grouping structure. First, the definition of the implicit state mixed linear regression model is presented. Thereafter, the Bayesian adaptive group Lasso method is used to determine the penalty function and parameters, after which each parameter's specific form of the fully conditional posterior distribution is calculated. Moreover, the Gibbs algorithm design is outlined. Simulation experiments are conducted to compare the variable selection and parameter estimation effects in different states. Finally, a dataset of Alzheimer's Disease is used for application analysis. The results demonstrate that the proposed method can identify the observation from different hidden states, but the results of the variable selection in different states are obviously different.

## Introduction

Multiple observation data of each index of the sample are required in biomedical and econometric research. Such data are usually referred to as longitudinal data. The mixed linear regression model is commonly used for fitting these data. In general, mixed linear regression models contain two parts: fixed effects and random effects that are subject to an unknown distribution. The variable selection problem in a mixed linear regression model usually focuses on the variable selection in the fixed effect part.

In recent years, the class of variable selection methods with penalty functions has become very popular. These methods are based on the least absolute shrinkage and selection operator (i.e., Lasso) method proposed by Tibshiran [1]. This class of penalty methods can perform variable selection and parameter estimation, and exhibits good stability and strong statistical properties. For example, the SCAD(Smoothly Clipped Absolute Deviation) penalty that was proposed by Fan and Li [2] can satisfy several excellent properties such as asymptotic unbiasedness, sparsity, and continuity. Zou [3] presented adaptive Lasso, which exhibits

108). This work was also supported by Yunnan Provincial Department of Education of China (Award Numbers: 2022J0810, 2023J1028). The funders had no role in study design, data collection and analysis, decision to publish, or preparation of the manuscript.

**Competing interests:** The authors have declared that no competing interests exist.

consistency when the number of variables is fixed and the sample size approaches infinity. This method solves the problem of poor consistency in Lasso estimation. Moreover, the adaptive group Lasso proposed by Wang and Leng [4] assigns different adjustment parameters to different regression system arrays, whereby effective variable selection and coefficient estimation can be performed, and subsequently, improved results can be obtained.

When Tibshiran proposed the Lasso method, he proved that when the prior distribution of the regression coefficient is a Laplace distribution, the estimation result of the regression coefficient that is obtained by the Lasso method is consistent with the result of the maximum a posteriori probability estimate, which led to the new concept of Bayesian Lasso. As the Bayesian method exhibits excellent stability and high computational efficiency, this method has been rapidly expanded. On this basis, Park and Casella [5] proposed a complete Bayesian model with conditional Laplace distribution as the prior distribution, and used Gibbs sampling to estimate the posterior distribution of the parameters. Subsequently, Kyung [6] further extended this model and proposed a complete Bayesian formula that can be combined with several variants of Lasso. Leng [7] extended this model to the complete Bayesian adaptive Lasso and applied it to the variable selection of linear models. Lykou [8] used the Bayesian Lasso method to select the model variables. Khondker [9] further extended this method to the Bayesian covariance Lasso method. Raman [10] proposed the Bayesian version of group Lasso method, applied it to contingency tables, and proved its stability and efficiency. Ibrahim [11] introduced the SCAD penalty and adaptive Lasso into the mixed linear regression model. Feng and Wang [12] presented the Bayesian adaptive group Lasso method and applied it to the semiparametric structural equation model. Kang and Song [13] applied the Bayesian adaptive group Lasso to the semiparametric hidden Markov model.

However, in general, the research on variable selection with a grouping structure of the explanatory variables under a mixed linear regression model with an implicit state remains lacking, and few studies have used Bayesian Lasso and its variants to solve this problem. In this study, we introduce the Bayesian adaptive group Lasso into the mixed linear regression model with hidden states to select the variables and estimate the parameters. The purpose is to explore the screening of explanatory variables in a mixed linear regression model when the samples have different states, and the explanatory variables are significant in some states and not significant in others.

The remainder of this paper is organized as follows: Section II introduces the basic form of the mixed linear regression model and its variable selection, Bayesian theory, Bayesian Lasso and its extension method, and the MCMC sampling algorithm, with a focus on the Bayesian adaptive group Lasso method. Section III introduces the core theory of this paper. First, the data and mixed linear regression model used in this study are outlined. Thereafter, the use of Bayesian adaptive group Lasso to estimate the parameters and select the variables under this mixed linear regression model is presented. Furthermore, the fully conditional posterior distribution of the unknown parameters involved in the Bayesian hierarchical model is derived. Finally, the specific algorithm steps of the Gibbs sampling in this study are provided. Section IV presents the application research. Subsequently, the estimation effect of the methods and algorithms on the real parameters and variable selection accuracy is valuated according to the numerical simulation results, and an example is provided. Section V summarizes the paper.

## Model description

Consider the following mixed linear regression model, where the observed individuals are recorded as $i = 1, 2, \cdots, N$, and the observed is $t = 1, 2, \cdots, T$. Under the condition $S_{it} = s$, the

regression model is:

$$y_{it} = \boldsymbol{x}_{it}\boldsymbol{\theta}_s + \boldsymbol{z}_{it}\boldsymbol{u}_s + \boldsymbol{\varepsilon}_{it}. \tag{1}$$

In the above, $\boldsymbol{\varepsilon}_{it}$ is the random error, which is independently and identically distributed in $N(0, \sigma^2)$, $S_{it}$ is the state of the $i$-th sample at the $t$-th observation, $S_{it} = s$ means that the model is defined in the specific state $s$. Parameter $\boldsymbol{\theta}_s$ is the unknown regression coefficient, which is also known as the fixed effect, and $\boldsymbol{\theta}_s = (\boldsymbol{\alpha}_s, \boldsymbol{\beta}_s)^T$. The explanatory variables corresponding to $\boldsymbol{\alpha}_s$ are independent of one another. $\boldsymbol{\beta}_s$ represents the coefficient corresponding to the explanatory variable with a grouping structure.

Furthermore, $\boldsymbol{\alpha}_s$ is an L-dimensional vector, $\boldsymbol{\beta}_s$ is a p-dimensional vector, and $x_{it} = (x_{it1}, x_{it2}, \cdots, x_{it(p+L)})$ is the known explanatory variable. Let the unknown random vector $\boldsymbol{u}_s$ be m-dimensional. Parameter $\boldsymbol{u}_s$ is often referred to as the random effect, and it is generally assumed that $\boldsymbol{u}_s \sim N_n(\boldsymbol{0}, \sigma^2_{u_s}\boldsymbol{I}_m)$. Thus, we know that vector $\mathbf{z}_{it}$ is m-dimensional, and because the model established in this study is red the longitudinal data model in the mixed linear regression model, $\mathbf{z}_{it}$ is a vector in which the $i$-th component is 1 and the other components are 0, $i = 1, 2, \cdots, N$.

For state $S_{it}$, the following assumption applies:

$$P(S_{it} = s) = q_s, \tag{2}$$

where $s = 1, 2, \cdots, S. q_s$ is an unknown constant value with $\sum_{s=1}^{S} q_s = 1$, and $S$ is a known positive integer; that is, the total number of states is known.

As the observation values in different states will not affect the theoretical form of the conditional prior of each unknown parameter and its corresponding full conditional posterior distribution, but only the specific numerical calculation, the distinction of the states of each observational value is discussed in the iterative calculation [14]. Therefore, the subsequent theoretical part is investigated in the specific state $s$. For convenience of the description, the subscript $s$ is omitted. In specific state $s$, the model is abbreviated as

$$y_{it} = \boldsymbol{x}_{it}\boldsymbol{\theta} + z_{it}\boldsymbol{u} + \boldsymbol{\varepsilon}_{it}, \tag{3}$$

where $\boldsymbol{\theta} = (\boldsymbol{\alpha}, \boldsymbol{\beta})^T$, $\boldsymbol{u}$ is distributed in $N(\boldsymbol{0}, \sigma^2_u \boldsymbol{I}_m)$, $\boldsymbol{\varepsilon}_{it}$ is distributed in $N(\boldsymbol{0}, \sigma^2)$, and $\boldsymbol{\alpha} = (\alpha_1, \alpha_2, \cdots, \alpha_L)^T$. All explanatory variables with a grouping structure are divided into J groups. The set of subscripts of each group is marked as $G_j, j = 1, 2, \cdots, J$. Thus, we can rewrite $\boldsymbol{\theta} = (\boldsymbol{\alpha}, \boldsymbol{\beta})^T$ as $\boldsymbol{\theta} = (\alpha_1, \alpha_2, \cdots, \alpha_L, \boldsymbol{\beta}_{G_1}, \boldsymbol{\beta}_{G_2}, \cdots, \boldsymbol{\beta}_{G_j})^T$.

## Bayesian inference principle

### Bayesian adaptive group Lasso

In this study, the Bayesian adaptive group Lasso has the following penalty function form:

$$p(\lambda, \gamma) = \sum_{l=1}^{L} \lambda_l |\alpha_l| + \sum_{j=1}^{J} \gamma_j ||\boldsymbol{\beta}_{G_j}||_{K_j}, \tag{4}$$

where $||\boldsymbol{\beta}_{G_j}||_{K_j} = (\boldsymbol{\beta}'_{G_j} K_j \boldsymbol{\beta}_{G_j})^{\frac{1}{2}}$, positive definite matrix $K_j = p_j \boldsymbol{I}_{p_j}$, $\boldsymbol{I}_{p_j}$ is a $p_j$-order identity matrix, and $\lambda_l$ and $\gamma_j$ are positive penalty parameters that have positive values. and $\gamma_j$ can be selected to calculate the corresponding full conditional posterior distribution, and the estimated value can be obtained by the Gibbs method [15].

We introduce the conditional Laplacian prior as the prior distribution of the coefficients of the explanatory variable [16], rewrite the model into a hierarchical structure, provide the fully

conditional posterior distribution of all the parameters to be estimated, and subsequently, calculate their estimated values according to Gibbs.

The conditional Laplace prior for coefficient $\boldsymbol{\alpha}$ is

$$p(\boldsymbol{\alpha}|\sigma^2) = \prod_{l=1}^{L} \frac{\lambda_l}{2\sigma} \exp(-\frac{\lambda_l|\alpha_l|}{\sigma}), \tag{5}$$

where $\alpha_l$ is the $l$-th component of $\boldsymbol{\alpha}$, which is independent and identically distributed in a univariate Laplace conditional distribution [17], with the location parameter 0 and scale parameter $\frac{\sigma}{\lambda_l}$.

The conditional Laplace prior for coefficient $\boldsymbol{\beta}$ is

$$p(\boldsymbol{\beta}|\sigma^2) \propto \prod_{j=1}^{J} \exp(-\frac{\gamma_j||\boldsymbol{\beta}_{G_j}||K_j}{\sigma}), \tag{6}$$

where $\boldsymbol{\beta}_{G_j}$, which denotes the components of $\boldsymbol{\beta}$, is independent and identically distributed in a multivariate Laplace distribution.

Subsequently, the above Laplace prior distribution is expressed as a normal mixed distribution with an exponential mixed distribution [18, 19]:

For $\boldsymbol{\alpha}$:

$$\frac{\lambda_l}{2\sigma} \exp(-\frac{\lambda_l|\alpha_l|}{\sigma}) = \int_0^{+\infty} \frac{1}{\sqrt{2\pi\sigma\tau}} \exp(-\frac{\alpha_l^2}{2\sigma^2\tau^2}) \frac{\lambda^2}{2} \exp(-\frac{\lambda^2}{2}\tau^2) d\tau. \tag{7}$$

For $\boldsymbol{\beta}$:

$$\exp(-\frac{\gamma_j||\boldsymbol{\beta}_{G_j}||K_j}{\sigma}) \propto \int_0^{+\infty} N_{p_j}(\mathbf{0}_{p_j}, \sigma^2\tau_{G_j}^2 K_j^{-1}) Gamma(\frac{p_{j+1}}{2}, \frac{\gamma_j^2}{2}) d\tau_{G_j}. \tag{8}$$

For convenience of the description, we combine the components of each parameter: let $\boldsymbol{\varepsilon} = (\varepsilon_{11}, \varepsilon_{12}, \cdots, \varepsilon_{mT})^T$ be an $mT$-dimensional vector, $\mathbf{Z} = (z_{11}, z_{12}, \cdots, z_{mT})^T$ be a matrix of $mT \times mT$, and $\Sigma = \sigma^2\mathbf{I}_{mT}$ be a matrix of $mT \times mT$. Moreover, $\boldsymbol{D} = \sigma_u^2\boldsymbol{I}_{mT}$ is a matrix of $m \times m$.

Let $\boldsymbol{\varepsilon}^* = \mathbf{Z}\boldsymbol{u} + \boldsymbol{\varepsilon}$, which is distributed in $N_{mT}(\mathbf{0}, \Sigma + \mathbf{Z}D\mathbf{Z}^T)$. Therefore, according to the model assumption, the conditional distribution of the explained variable $Y$ can be obtained as follows:

$$Y|\boldsymbol{\theta}, \Sigma, \boldsymbol{D} \sim N_{mT}(\boldsymbol{X}\boldsymbol{\theta}, \Sigma + \mathbf{Z}D\mathbf{Z}^T). \tag{9}$$

Let $\Sigma + ZDZ^T \approx \sigma^{*2}I_{mT}$, we can rewrite (9) as follows:

$$Y|\boldsymbol{\theta}, \Sigma, \boldsymbol{D} \sim N_{mT}(\boldsymbol{X}\boldsymbol{\theta}, \sigma^{*2}\boldsymbol{I}_{mT}), \tag{10}$$

where the $^*$ of $\sigma^{*2}\boldsymbol{I}_{mT}$ is omitted for a succinct description. The prior for parameter $\sigma^2$ is set as the inverse gamma distribution. Thus, the model can be expressed as the following hierarchical

model:

$$Y|X, \boldsymbol{\theta}, \sigma^2 \sim N_{mT}(X\boldsymbol{\theta}, \sigma^2 I_{mT}),$$

$$\boldsymbol{\alpha}|\sigma^2, \tau_{\alpha_1}^2, \tau_{\alpha_2}^2, \cdots, \tau_{\alpha_L}^2 \sim N_L(\mathbf{0}_L, \sigma^2 \boldsymbol{\Sigma}_\alpha),$$

$$\boldsymbol{\Sigma}_\alpha = diag(\tau_{\alpha_1}^2, \tau_{\alpha_2}^2, \cdots, \tau_{\alpha_L}^2),$$

$$\tau_{\alpha_l}^2|\lambda_l^2 \sim Gamma\left(1, \frac{\lambda_l^2}{2}\right),$$

$$\boldsymbol{\beta}_{G_j}|\sigma^2, \tau_{G_j}^2 \sim N_{p_j}(\mathbf{0}_{p_j}, \sigma^2 \tau_{G_j}^2 K_j^{-1}), \tag{11}$$

$$\tau_{G_j}^2|\gamma_j^2 \sim Gamma\left(\frac{p_{j+1}}{2}, \frac{\gamma_j^2}{2}\right),$$

$$\sigma^2 \sim In - Gamma(a, b)$$

$$\lambda_l^2 \sim Gamma(a_\lambda, b_\lambda)$$

$$\gamma_j^2 \sim Gamma(a_\gamma, b_\gamma)$$

where $a$, $b$, $a_\lambda$, $b_\lambda$, $a_\gamma$ and $b_\gamma$ are hyperparameters.

## Gibbs sampling

The hierarchical model for Bayesian adaptive group Lasso was obtained in the previous section. It is necessary to solve the fully conditional posterior distribution of all unknown parameters to use Gibbs sampling to estimate the parameters involved in the model [20, 21].

According to the hierarchical model, all conditional posterior distributions of the parameters are obtained as follows:

$$p(\lambda_l^2|\cdot) \sim Gamma(\alpha_\lambda + 1, b_\lambda + \frac{\tau_{\alpha_l}^2}{2}),$$

$$p(\gamma_j^2|\cdot) \sim Gamma(\alpha_\lambda + \frac{p_{j+1}}{2}, b_\gamma + \frac{\tau_{G_j}^2}{2}),$$

$$p(\tau_{\alpha_l}^{-2}|\cdot) \sim Gamma(\sqrt{\frac{\lambda_l^2 \sigma^2}{|\alpha_l|^2}}, \lambda_l^2),$$

$$p(\tau_{G_j}^{-2}|\cdot) \sim Gamma(\sqrt{\frac{\gamma_j^2 \sigma^2}{||\beta_{G_j}||K_l^2}}, \gamma_j^2),$$

$$p(\boldsymbol{\alpha}|\cdot) \sim N(\boldsymbol{A}^{-1}\boldsymbol{X}_\alpha^T\boldsymbol{Y}_\alpha, \sigma^2\boldsymbol{A}^{-1}),$$

$$p(\boldsymbol{\beta}_{G_j}|\cdot) \sim N(\boldsymbol{B}^{-1}\boldsymbol{X}_{\beta_{G_j}}^T\boldsymbol{Y}_{\beta_{G_j}}, \sigma^2\boldsymbol{B}^{-1}),$$

$$(\sigma^{-2}|\cdot) \sim Gamma(\frac{\Sigma p_j + mt + L}{2}, b + \frac{1}{2}(\boldsymbol{Y}-\boldsymbol{X}\boldsymbol{\theta})^T(\boldsymbol{Y}-\boldsymbol{X}\boldsymbol{\theta}) + \boldsymbol{\alpha}^T\boldsymbol{\Sigma}_\alpha^{-1}\boldsymbol{\alpha} + \Sigma\boldsymbol{\beta}_{G_j}^T\tau_{G_j}^{-2}\boldsymbol{K}_j\boldsymbol{\beta}G_j),$$

where $\boldsymbol{A} = \boldsymbol{X}_\alpha^T\boldsymbol{X}_\alpha + \boldsymbol{\Sigma}_\alpha^{-1}, \boldsymbol{Y}_\alpha = \boldsymbol{Y} - \boldsymbol{X}_\alpha\boldsymbol{\beta}, \boldsymbol{B} = \boldsymbol{X}_{\beta_{G_j}}^T\boldsymbol{X}_{\beta_{G_j}} + \tau_{G_j}^{-2}\boldsymbol{K}_j, \boldsymbol{Y}_{\beta_{G_j}} = \boldsymbol{Y} - \boldsymbol{X}_{(\beta_{G_j})}\boldsymbol{\theta}_{(\beta_{G_j})}.$

Gibbs sampling can be used for parameter estimation once all of the conditional posterior distributions of all unknown parameters have been obtained. The confidence interval criterion method proposed by Li and Lin [17] is used for the variable selection. According to this method, for the coefficients corresponding to variables without a grouping structure $\boldsymbol{\alpha}$, if the 95% confidence interval does not cover zero, the variable can be considered as significant; otherwise, the variable is considered as not significant and is eliminated. For the coefficients corresponding to variables with a grouping structure $\boldsymbol{\beta}$, if the 95% confidence interval of the estimated coefficient of a variable in the group covers zero, the entire group of variables is eliminated.

In the Gibbs process, the specific iteration procedure is as follows:

(1) The specific state of the observed value is unknown but the total number of hidden states is known, and an initial value is assigned to the hidden states:

$$S_{it}^{(0)} = s, i = 1, 2, \cdots, n; t = 1, 2, \cdots, T; s = 1, 2, \cdots, S.$$

Let $\boldsymbol{\zeta} = (\boldsymbol{\theta}, \sigma^{-2}, \boldsymbol{\tau}_\alpha^{-2}, \boldsymbol{\lambda}^2, \boldsymbol{\tau}_\beta^{-2}, \boldsymbol{\gamma}^2) = (\boldsymbol{\zeta}_1, \boldsymbol{\zeta}_2, \cdots, \boldsymbol{\zeta}_6)$. The initial value of each parameter under specific state $s$ is:

$$\boldsymbol{\zeta}^{(0)} = (\boldsymbol{\theta}^{(0)}, (\sigma^{-2})^{(0)}, (\boldsymbol{\tau}_\alpha^{-2})^{(0)}, (\boldsymbol{\lambda}^2)^{(0)}, (\boldsymbol{\tau}_\beta^{-2})^{(0)}, (\boldsymbol{\gamma}^2)^{(0)}).$$

(2) For the $k$-th iteration:
sample the parameters in each state $s$, $s = 1, 2, \cdots, S$:
sample $\boldsymbol{\zeta}_1^{(k)}|_s$ from $p(\boldsymbol{\zeta}_1|\boldsymbol{\zeta}_2^{(k-1)}, \cdots, \boldsymbol{\zeta}_6^{(k-1)})|_s$,
sample $\boldsymbol{\zeta}_2^{(k)}|_s$ from $p(\boldsymbol{\zeta}_2|\boldsymbol{\zeta}_1^{(k-1)}, \boldsymbol{\zeta}_3^{(k-1)}, \cdots, \boldsymbol{\zeta}_6^{(k-1)})|_s$,
$\cdots\cdots$

sample $\zeta_6^{(k)}|_s$ from $p(\zeta_6|\zeta_1^{(k-1)}, \cdots, \zeta_5^{(k-1)})|_s$,

until all parameters in all states have been converged.

Subsequently, the extracted parameters are used to calculate the full conditional probability density function of the hidden state:

$$p(S_{it}^{(k)} = s|x_{it}, y_{it}, \zeta_s^{(k)}) = \frac{q_s^{(k-1)} p_s(x_{it}, y_{it}|\zeta_s^{(k)})}{\sum_{s=1}^{S} q_s^{(k-1)} p_s(x_{it}, y_{it}|\zeta_s^{(k)})},$$

where $q_s^{(k-1)} = p(S_{it}^{(k-1)} = s)$ and $p_s(x_{it}, y_{it}|\zeta_s^{(k)})$ is the likelihood function of the observation in state $s$. Thus, the conditional probability density function of all hidden states is obtained, following which the state of each observation at this time can be obtained using distribution $U(0, 1)$ as auxiliary sampling:

$$S_{it}^{(k)} = s, s = 1, 2, \cdots, S.$$

Update parameter $q_s$:

$$q_s^{(k)} = p(S_{it}^{(k)} = s)$$

The $k$-th iteration ends.

(3) Return to step (2) and perform the $(k + 1)$-th iteration until the target number of iterations is reached.

## Simulation experiment

### Model settings

The main purpose of the numerical simulations are testing the accuracy of the model parameter estimation and variable selection, the accuracy of determining the state of each observation, and the differences in the variable selection results under different states. Moreover, the effects of different sample sizes on the parameter estimation were investigated.

(1) The simulation settings are as follows:

A total of 100 experiments were conducted in which the following situations were considered each time: observation times $T = 3$; sample size $N = 100$, and 300; number of hidden states $S = 2$. The probability that each observation value belonged to state 1 or state 2 was the same, namely 0.5.

In the first state: $\boldsymbol{\alpha} = (-1.7, 1.3), \boldsymbol{\beta} = (\boldsymbol{\beta}_{G_1}, \boldsymbol{\beta}_{G_2}, \boldsymbol{\beta}_{G_3})$, and $\boldsymbol{\beta}_{G_1} = (2.0, -0.4, 1.3)$; $\boldsymbol{\beta}_{G_2} = (0.6, 0, -1.1); \boldsymbol{\beta}_{G_3} = (0, 0, 0)$.

In the second state: $\boldsymbol{\alpha} = (0.3, -1.2), \boldsymbol{\beta} = (\boldsymbol{\beta}_{G_1}, \boldsymbol{\beta}_{G_2}, \boldsymbol{\beta}_{G_3})$, and $\boldsymbol{\beta}_{G_1} = (0, 0, 0); \boldsymbol{\beta}_{G_2} = (0, 0, 0); \boldsymbol{\beta}_{G_3} = (-1.5, 2.2, 0.5)$.

The settings in the two states were considered for design matrix $\boldsymbol{X}$:

The part corresponding to coefficient $\boldsymbol{\alpha}$, namely $\boldsymbol{X_\alpha}$, was distributed in the multivariate normal $N(\boldsymbol{0}, \boldsymbol{I})$, where $\boldsymbol{I}$ is the identity matrix.

The part corresponding to coefficient $\boldsymbol{\beta} = (\boldsymbol{\beta}_{G_1}, \boldsymbol{\beta}_{G_2}, \boldsymbol{\beta}_{G_3})$, namely $\boldsymbol{X}_{\boldsymbol{\beta}_{G_1}}, \boldsymbol{X}_{\boldsymbol{\beta}_{G_2}}, \boldsymbol{X}_{\boldsymbol{\beta}_{G_3}}$, was distributed in $N(\boldsymbol{0}, \boldsymbol{\Sigma}_{\boldsymbol{\beta}_{G_1}}), N(\boldsymbol{0}, \boldsymbol{\Sigma}_{\boldsymbol{\beta}_{G_2}})), N(\boldsymbol{0}, \boldsymbol{\Sigma}_{\boldsymbol{\beta}_{G_3}})$.

As there was a strong correlation between the components of $\boldsymbol{\beta}_{G_j}, j = 1, 2, 3$, the following settings were available: the element in row $i$ and column $k$ of $\boldsymbol{\Sigma}_{\boldsymbol{\beta}_{G_1}}$ was $0.7^{|i-k|}$, $i, k = 1, 2, 3$; the element in row $i$ and column $k$ of $\boldsymbol{\Sigma}_{\boldsymbol{\beta}_{G_2}}$ was $0.6^{|i-k|}$, $i, k = 1, 2, 3$; the element in row $i$ and column $k$ of $\boldsymbol{\Sigma}_{\boldsymbol{\beta}_{G_3}}$ was $0.4^{|i-k|}$, $i, k = 1, 2, 3$.

The following settings were used for the random effects:

$$u_i \sim N(0,1), i = 1, 2, \cdots, n, \varepsilon_{ij} \sim N(0,1), i = 1, 2, \cdots, n, j = 1, 2, 3.$$

(2) Hyperparameter and MCMC settings:

Hyperparameters $a$, $b$, $a_\lambda$, $b_\lambda$, $a_\gamma$, and $b_\gamma$ in the hierarchical model (11) were set to [22]:(prior I) $a = 1$, $b = 0.1$, $a_\lambda = 1$, $b_\lambda = 0.1$, $a_\gamma = 1$, and $b_\gamma = 0.01$.

The number of iterations of the MCMC was set to 5000. Three groups of different initial values were set for all parameters to be estimated and the EPSR values of three parallel simulation sequences of all parameters were calculated. When the number of iterations was 2000, the EPSR values of all parameters were less than 1.2. This indicated that the sample converged at 2000 iterations. Therefore, the samples that were obtained from the first 2500 iterations were removed as aging values and only the following 2500 data were retained for analysis to ensure convergence. The posterior mean was used as the estimated value of each parameter.

## Analysis of results

After repeating the experiment 100 times, the estimated conditions of each coefficient selected in the model could be classified in two different quantities and two states, as indicated in Table 1.

It can be observed from Table 1 that the model generally had a good estimation effect for each parameter, and the estimation effect on each component of $\boldsymbol{\alpha}$ was better than that on each component of $\boldsymbol{\beta}_{G_j}$. This may be because the three components of $\boldsymbol{\beta}_{G_j}$ had a strong correlation with one another and imposed penalties on the entire group, so the estimation effect of a single component was somewhat poor.

Furthermore, the accuracy of the parameter estimation increased with the increase in the sample size, indicating that the estimation effect of the model increased.

The model variable selection was also investigated, and the 95% confidence interval of the posterior mean value of each parameter was calculated. For parameter $\boldsymbol{\alpha}$, if the confidence interval of its components was covered to zero, the corresponding variables were removed. For

**Table 1. Estimate results under prior I.**

| | n = 100 | | | | n = 300 | | | |
| | s = 1 | | s = 2 | | s = 1 | | s = 2 | |
| Par | Bias | RMSE | Bias | RMSE | Bias | RMSE | Bias | RMSE |
|---|---|---|---|---|---|---|---|---|
| $\boldsymbol{\alpha}$ | 0.061 | 0.094 | 0.039 | 0.068 | 0.024 | 0.063 | 0.017 | 0.045 |
| | -0.052 | 0.085 | 0.047 | 0.064 | -0.015 | 0.071 | 0.026 | 0.038 |
| $\boldsymbol{\beta}_{G_1}$ | -0.048 | 0.091 | -0.025 | 0.102 | -0.032 | 0.063 | -0.011 | 0.082 |
| | -0.048 | 0.091 | -0.025 | 0.102 | -0.032 | 0.063 | -0.016 | 0.093 |
| | 0.035 | 0.114 | -0.017 | 0.086 | 0.019 | 0.094 | 0.014 | 0.064 |
| $\boldsymbol{\beta}_{G_2}$ | -0.034 | 0.134 | -0.072 | 0.094 | -0.027 | 0.102 | -0.039 | 0.080 |
| | 0.028 | 0.097 | -0.126 | 0.091 | 0.016 | 0.075 | -0.047 | 0.065 |
| | 0.019 | 0.062 | 0.047 | 0.112 | 0.012 | 0.044 | -0.029 | 0.083 |
| $\boldsymbol{\beta}_{G_3}$ | 0.052 | 0.114 | -0.035 | 0.097 | 0.039 | 0.085 | -0.023 | 0.083 |
| | 0.031 | 0.086 | -0.027 | 0.074 | 0.018 | 0.064 | -0.019 | 0.063 |
| | 0.074 | 0.092 | 0.059 | 0.112 | 0.046 | 0.077 | 0.038 | 0.087 |

RMSE:root mean square error

**Table 2. Identification results of insignificant variables.**

| State | Sample size | Corr | Incorr |
|-------|-------------|------|--------|
| $S = 1$ | $n = 100$ | 1 | 1 |
|       | $n = 300$ | 1 | 1 |
| $S = 2$ | $n = 100$ | 2 | 0 |
|       | $n = 300$ | 2 | 0 |

Corr: average number of correct zeros; Incorr: average number of incorrect zeros.

parameters $(\boldsymbol{\beta}_{G_1}, \boldsymbol{\beta}_{G_2}, \boldsymbol{\beta}_{G_3})$, if the confidence interval of the components of $\boldsymbol{\beta}_{G_j}, j = 1, 2, 3$ was covered to zero, the entire group of $X_{\boldsymbol{\beta}}G_1$ was removed.

In this study, the components $\boldsymbol{\beta}_{G_j}, j = 1, 2, 3$ of $\boldsymbol{\beta}$ were considered as a whole. In each experiment, among the three components of the estimated value of $\boldsymbol{\beta}$ in two states, the true value was zero, and the variable selection result was the number of components excluding their corresponding variables. A total of 100 record results were obtained in each of the two states, and the mean value of the results was the average of the correct zeros in Table 2. Accordingly, the average of incorrect zeros indicated that the true value was not zero, but the variable selection result was the average of the number of coefficient components, excluding their corresponding variables.

It can be observed from Table 3 that in the 100 repeated experiments, the two group vectors with zero real coefficients in state 2 were eliminated. However, in state 1, only one group vector had a true coefficient of zero, but two group vectors were excluded; that is, one group vector with a true coefficient of non-zero was excluded from the model.

According to Tables 1–3, only one of the components of parameter $\boldsymbol{\beta}_{G_2}$ in state 1 had all zero values. However, according to the "confidence interval criterion", if the confidence interval of a component covers zero, the entire corresponding group of variables should be eliminated. In $\boldsymbol{\beta}_{G_2}$, as only one component had a large coefficient, it was considered that $X_{\boldsymbol{\beta}_{G_2}}$ as a whole was not significant and needed to be eliminated.

## Sensitivity analysis

In this section, we conduct sensitivity analysis to examine whether the proposed method is sensitive to the prior specification. we reset the hyperparameters as follows:(priorII) $a = 6$, $b = 4$, $a_\lambda = 2$, $b_\lambda = 0.01$, $a_\gamma = 3$, $b_\lambda = 0.05$. The MCMC setting is not changed. Table 4 is the parameter estimate result under prior II.

The estimated results of parameters in Table 4 are similar to those in Table 1. The experimental results show that the proposed variable selection method is robust to the prior distribution hyperparameters.

**Table 3. Frequency of group variables selected in model.**

| State | Sample size | $X_{\beta_{G_1}}$ | $X_{\beta_{G_2}}$ | $X_{\beta_{G_3}}$ |
|-------|-------------|-------------------|-------------------|-------------------|
| $S = 1$ | $n = 100$ | 100 | 0 | 0 |
|       | $n = 300$ | 100 | 0 | 0 |
| $S = 2$ | $n = 100$ | 0 | 0 | 100 |
|       | $n = 300$ | 0 | 0 | 100 |

**Table 4. Estimate results under prior II.**

| Par | n = 100 | | | | n = 300 | | | |
|---|---|---|---|---|---|---|---|---|
| | s = 1 | | s = 2 | | s = 1 | | s = 2 | |
| | Bias | RMSE | Bias | RMSE | Bias | RMSE | Bias | RMSE |
| $\alpha$ | 0.058 | 0.088 | 0.0045 | 0.084 | -0.018 | 0.055 | 0.021 | 0.054 |
| | -0.067 | 0.082 | 0.039 | 0.047 | -0.009 | 0.068 | -0.013 | 0.042 |
| $\beta_{G_1}$ | -0.042 | 0.086 | 0.042 | 0.094 | -0.027 | 0.072 | 0.020 | 0.093 |
| | 0.025 | 0.095 | 0.019 | 0.073 | 0.016 | 0.084 | 0.015 | 0.104 |
| | 0.051 | 0.073 | -0.025 | 0.061 | 0.011 | 0.075 | 0.018 | 0.087 |
| $\beta_{G_2}$ | 0.042 | 0.103 | -0.065 | 0.084 | -0.022 | 0.090 | 0.015 | 0.055 |
| | 0.017 | 0.079 | -0.083 | 0.086 | 0.015 | 0.084 | -0.037 | 0.063 |
| | -0.035 | 0.081 | 0.061 | 0.101 | 0.019 | 0.063 | 0.021 | 0.071 |
| $\beta_{G_3}$ | -0.049 | 0.089 | 0.041 | 0.104 | -0.027 | 0.059 | 0.018 | 0.068 |
| | 0.036 | 0.069 | 0.025 | 0.070 | 0.024 | 0.071 | -0.014 | 0.049 |
| | 0.064 | 0.095 | 0.041 | 0.083 | -0.031 | 0.067 | 0.036 | 0.074 |

## Case study

The proposed model and method can be used in the study of Alzheimer's disease to illustrate the practicability of the model and method. The data and more information can be found on its website(www.adni-info.org). Because many individuals had information missing, this research did not consider the missing data problem, we deleted individuals with information missing. So we selected 512 patients and collected their clinical information and basic variables at the base period, 6 months, 12 months, 24 months and 36 months. $N = 512$ and $T = 5$ in this model. The specific information of response variables and interest covariables initially selected by us is shown in Table 5.

In the model, FAQ(Functional Assessment Qestionaire) scores were selected as the response variable($y_{it}$) to reflect the cognitive and behavioral abilities of respondents. Among the 11 possible interest variables, $X_1, X_2, X_3, X_4$ is the inborn and unchangeable biological genetic information, $X_5, X_6, X_7$ is the changeable biological information, $X_8, X_9$ is the past historical information, and $X_{10}, X_{11}$ is the current social attribute that may change. Therefore, we

**Table 5. Variable details.**

| Variable | Name | Value range |
|---|---|---|
| $Y$ | FAQ Sore | [0, 30] |
| $X_1$ | Gender | Man:1 Woman:0 |
| $X_2$ | Race | White:1 Else:0 |
| $X_3$ | APOE-$\varepsilon4$ | Yes:1 No:0 |
| $X_4$ | Family history | Yes:1 No:0 |
| $X_5$ | Age | $\geq 0$ |
| $X_6$ | The proportion of amyloid $\beta$ in cerebral and renal fluid($A\beta_{42}$) | $0 \sim 100\%$ |
| $X_7$ | The proportion of brain volume in the hippocampus | $0 \sim 100\%$ |
| $X_8$ | Years of education | $\geq 0$ |
| $X_9$ | Head trauma | Yes:1 No:0 |
| $X_{10}$ | Spouse | Yes:1 No:0 |
| $X_{11}$ | Working | Yes:1 No:0 |

divide 11 variables into 4 groups:$X_{\beta_{G_1}} = c(X_1, X_2, X_3, X_4)$,$X_{\beta_{G_2}} = c(X_5, X_6, X_7)$,$X_{\beta_{G_3}} = c(X_8, X_9)$,$X_{\beta_{G_4}} = c(X_{10}, X_{11})$. We roughly divide the respondents into two states, one is with cognitive and behavioral disorders, and the other is without or with slight cognitive and behavioral disorders. We need to study the following issues: 1.What are the factors that affect cognitive and behavioral abilities in each state? 2.In each state, what is the influence relationship between covariates and response variables?

The above problem is to select variables and estimate the parameters of model $Y = \alpha + X_{\beta_{G_1}} + X_{\beta_{G_2}} + X_{\beta_{G_3}} + X_{\beta_{G_4}} + \varepsilon$. Here $\alpha$ is the intercept term.

Before the empirical analysis, we first standardized the three variables, FAQ sore($y$), age ($X_5$) and years of education ($X_8$). We choose the hyperparameter of prior I in the analysis. Table 6 shows the results of variable selection and parameter estimation.

From Table 6, we can see the variable selection consequences of the model. Under state 1, three groups of variables $G_1, G_2, G_4$ have significant effects on response variables, while $G_3$ has no significant effects on response variables. Under state 2, $G_2$ has a significant impact on the response variable, while $G_1, G_3, G_4$ has no significant impact on the response variable.

Substituting the corresponding coefficients and variables into the model, we can get that when the respondents have cognitive impairment, the influence relationship model of FAQ score is $y = 1.71 - 0.25X_1 - 0.16X_2 + 0.72X_3 + 0.08X_4 + 0.11X_5 + 0.46X_6 - 0.29X_7 - 0.69X_{10} - 0.39X_{11}$. When the respondents has no cognitive impairment or mild cognitive impairment, the influence relationship model of FAQ score is $y = -0.84 + 0.27X_5 + 1.14X_6 - 0.43X_7$.

The results of variable selection show that different contents affect FAQ scores in different cognitive states. When the respondent is in a cognitive disorder state, the innate genetic information, changeable biological information and current social attributes will affect the respondent's cognitive ability (FAQ score). When the respondent is in the state of no cognitive impairment or mild cognitive impairment, only changeable biological information has a significant impact on cognitive ability (FAQ score).

From the consequences of variable selection, we found that the changeable biological information ($X_5, X_6, X_7$) had a significant impact on cognitive ability no matter the respondents were in any state. Further analysis shows that $X_5$ and $X_6$ have a positive impact on the FAQ score, and $X_7$ has a negative impact on the FAQ score. This shows that the older you get, the bigger $A\beta_{42}$ you get, the weaker your cognitive ability is. The larger the volume of

**Table 6. Estimate results of Alzheimer's disease data.**

| Paramete | State1 | State2 |
|---|---|---|
| $\alpha$ | 1.71 | -0.84 |
| $X_{\beta_{G_1}}$ | -0.25 | — |
| | -0.16 | — |
| | 0.72 | — |
| | 0.08 | — |
| $X_{\beta_{G_2}}$ | 0.11 | 0.27 |
| | 0.46 | 1.14 |
| | -0.29 | -0.43 |
| $X_{\beta_{G_3}}$ | — | — |
| | — | — |
| $X_{\beta_{G_4}}$ | -0.69 | — |
| | -0.39 | — |
| $\sigma^2$ | 1.29 | 0.64 |

hippocampus, the stronger the cognitive ability. In addition, under the condition of no cognitive impairment or slight cognitive impairment, the influence of innate genetic information $(X_1, X_2, X_3, X_4)$ and current social attributes $(X_{10}, X_{11})$ on cognitive ability is not significant. In the state of cognitive impairment, the influence of these two groups of variables on cognitive ability is significant. This is an interesting discovery. For example, does this mean that people of different genders have different risks of cognitive impairment? These results provide a novel perspective that deserves further inves.

In addition, the original data set gives the diagnostic status of each respondent at each test. We use the results of the last iteration of MCMC as the model to classify the status of respondents. Through comparison, out of 2048 sample points ($513 \times 4 = 2048$), 1962 sample points have positive classification results, with a correct rate of 95.8%. This shows that our model has good adaptability to data sets.

## Conclusions

In this study, Bayesian adaptive group Lasso was applied to the mixed linear regression model with hidden states, adaptive Lasso was applied to certain independent variables, and adaptive group Lasso was applied to several variables with a grouping structure. Under the Bayesian framework, the selection of the penalty function and penalty parameters as well as that of the prior distribution of each parameter was provided, following which the concrete form of all conditional posterior distributions of each parameter were calculated. The specific implementation steps of the Gibbs sampling were presented. Finally, the effects of the model parameter estimation and variable selection were discussed. The simulation analysis demonstrated that the proposed model can better identify the insignificant variables, eliminate the insignificant variables with a grouping structure, and estimate the parameters accurately. The case study verified that the same set of variables may or may not be significant in different states.

## Supporting information

**S1 File.**
(RAR)

## Author Contributions

**Conceptualization:** Hefei Liu.

**Data curation:** Rubing Li.

**Formal analysis:** Hefei Liu.

**Funding acquisition:** Yong Li, Hefei Liu.

**Methodology:** Hefei Liu.

**Project administration:** Yong Li.

**Software:** Yong Li.

**Writing – original draft:** Hefei Liu.

**Writing – review & editing:** Rubing Li.

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
