## [Decision Letter · Decision Letter 0]

17 Oct 2022

PONE-D-22-18597Study of Bayesian variable selection method on linear mixed regression modelsPLOS ONE

Dear Dr. Li,

Thank you for submitting your manuscript to PLOS ONE. After careful consideration, we feel that it has merit but does not fully meet PLOS ONE’s publication criteria as it currently stands. Therefore, we invite you to submit a revised version of the manuscript that addresses the points raised during the review process. Please submit your revised manuscript by Dec 01 2022 11:59PM. If you will need more time than this to complete your revisions, please reply to this message or contact the journal office at plosone@plos.org. Please include the following items when submitting your revised manuscript:A rebuttal letter that responds to each point raised by the academic editor and reviewer(s). You should upload this letter as a separate file labeled 'Response to Reviewers'.A marked-up copy of your manuscript that highlights changes made to the original version. You should upload this as a separate file labeled 'Revised Manuscript with Track Changes'.An unmarked version of your revised paper without tracked changes. You should upload this as a separate file labeled 'Manuscript'.

We look forward to receiving your revised manuscript.

Kind regards,

Lei Shi

Academic Editor

PLOS ONE

Journal Requirements:

"Specify the role(s) played."

3. Please update your submission to use the PLOS LaTeX template. The template and more information on our requirements for LaTeX submissions can be found at http://journals.plos.org/plosone/s/latex

Reviewers' comments:

Reviewer's Responses to Questions

**Comments to the Author**

1. Is the manuscript technically sound, and do the data support the conclusions?

Reviewer #1: Partly

Reviewer #2: Yes

2. Has the statistical analysis been performed appropriately and rigorously? 

Reviewer #1: No

Reviewer #2: Yes

3. Have the authors made all data underlying the findings in their manuscript fully available?

Reviewer #1: Yes

Reviewer #2: Yes

4. Is the manuscript presented in an intelligible fashion and written in standard English?

Reviewer #1: Yes

Reviewer #2: Yes

5. Review Comments to the Author

Reviewer #1: The submitted manuscript investigates the Bayesian variable selection problem in the context of linear mixed model with implicit state. The LASSO-type penalty term is employed as variable selection instrument and a Gibbs-type algorithm based on Laplacian-prior is proposed to implement the method. Some numerical studies are conducted to illustrate the performance of the proposed variable selection instrument. Variable selection (or more generally the model selection) is always the first problem needs to be answered in data analysis and investigating the Bayesian variable selection in the domain of linear mixed models is an important issue. However, after reading the manuscript, I feel that the research problem has not been investigated thoroughly. In specific, I have some concerns on the hyperparameter selection, design of simulation study and also the case study. Please see my specific comments in the attachment.

Reviewer #2: This paper introduces the Bayesian adaptive group Lasso method to investigate variable selection for the mixed linear regression model with an implicit state and explanatory variables with a grouping structure. The topic is interesting and the paper is well-written.

Comments:

1. Page 5, lines 5-6: Why is [yit|Sit = s] the observed time? What does the brackets mean?

2. An important aspect of any Bayesian analysis is when we use (close to or) noninformative

priors, posterior distributions can be improper (see, Hobert and Casella, 1996). Hence, it is

also important to have a complete sensitivity study about the choice of the hyper-parameters

for the priors.

3. For reproducibility purposes, I would suggest to make the code available (if it is possible) either

as supplementary material to be published online or as a reference to a gitHub website.

Minor Comments:

1. Pages 9-10: It seems unnecessary to replace (θ, σ−2, · · ·) with (ζ1, · · · , ζ6).

2. For comparison purpose, could you combine Table 1-3 and redesign the table?

References

[1] Hobert, J. P., Casella, G. (1996). The effect of improper priors on Gibbs sampling in hierarchical

linear mixed models. Journal of the American Statistical Association, 91(436), 1461-1473.

6. PLOS authors have the option to publish the peer review history of their article (what does this mean?). If published, this will include your full peer review and any attached files.

Reviewer #1: No

Reviewer #2: No

---

## [Author Response · Author response to Decision Letter 0]

22 Nov 2022

RE: “Study of Bayesian variable selection method on mixed linear regression models”

We are very grateful to the Editor, Associate Editor, and two reviewers for their constructive comments and suggestions, which have helped greatly in improving our paper. Our point-by-point responses are given below.

Response to Reviewer 1

1. Bayesian variable selction provides a promising way to inerpret the underlying mechanism of LASSO method. However, the Bayesian framework also induces further difficulties and one of the most important issues is how to determine the hyperparameters. In current study, very unfortunately, the authors failed to provide any details on how to determine hyperparameters in their proposed method (they just simply mentioned the value of hyperparameters used in the simulation). There are a number of immediate problems. Whether they are prefixed constants satisfy certain conditions? If so, any theoretical reasons? Or being selected from a set of candidate values based on a datadriven criterion? How to avoid overtraining of

overfitting? I want to see some in-depth theoretical and numerical investigations about these issues in the revised version.

Response: We are very grateful to the reviewer for the valuable comments. In order to observe the influence of the value of the hyperparameter on the model inference, we added the hyperparameter sensitivity test, and the experimental results of the two groups of hyperparameters were close.This shows that the inference method proposed in this paper is insensitive to hyperparameters.

Therefore, it is only necessary to select some non-information priors when setting

the hyperparameters of prior distribution. Please refer to page 11 of the revised version.

2. In pinciple, there are two typical setings for model selection i.e. the ”true -model world” where the trued data generating process (DGP)is nested within the candidate model set,and the ”non-true model world where DGP are unknown (Fynn et al, 2013) The curent manuscript studies the first setting. In linear models, Flynn et al. (2013) show that LASSO provides efficient model selection results. Then, another 

 mmediate question is that in the ”non-true model world”, whether the proposed model selection instrument can also possess similar properties?

Response:Thank you for the insightful point.You are right.We studied the first setting.To be honest,we haven’t considered the second case.After you asked this question,we had some thoughts. Maybe we will do some word in this field in the future .However,due to the immature

consideration,we dare not answer this question randomly.Please forgive us.

3. In the simulation study, the authors imposed a very strong signal-to-noise setting, where non-zero coefite are large in general. l’d like to see some ievetgaions on relaive weak signal settings.

Response: Thank you for your valuable question. We have reduced the non-zero coefficient and resimulation experiment according to your suggestion. Please refer to page 8 of the revised version.

4. The case study did not provide too much information! Is there any scientific evidence showing that the identifed variables are reasonable or not? Whether the selected model yields good prediction performance or not? Please consider this carefull! 

Response:We are very greatful to the reviewer for the valuable comment. We use the method to analysis a new Alzheimer’s disease data. With the increase of humen life expectancy, there are more and more patients with Alzheimer’s disease,so the research on Alzheimer’s disease is more and more important. Varible selection can screan out the factor that may affect Alzheimer’s disease, which is vary meaningful. We studied the Alzheimer’s disease dataset with the proposed model and method, and analyzed the results of vasiable selection and parameter estimation. Please refer to page 12 of the revised version.

5. Tables are separated on differrent pages and you can simply use floating tables to avoid this.

Response:Thank you for raising the good point. We deleted Table 2 and redesigned the table to merge Table 3 with Table 1. Please refer to page 9 of the revised version.

Response to Reviewer 2

1. Page 5, lines 5-6: Why is [yit|Sit = s]the observed time? What does the brackets meau?

Response: Sorry, we left out something in the original text. The revised paper has added “observe time is t = 1, 2, · · · T”. In the original paper, we want to use the brackets to represent the conditional distribution, that is, the y distribution of the observing variables in the state of s. The revised paper uses a more rigorous expression. Please refer to page 3 of the revised version.

2. An important aspect of any Bayesian analysis is when we use (close to or) noninformative priors, posterior distributions can be improper (see, Hobert and Casella, 1996). Hence, it is also important to have a complete sensitivity study about the choice of the hyper-parameters for the priors.

Response: We are very grateful to the reviewer for the valuable comments.In order to observe the influence of the value of the hyperparameter on the model inference, we added the hyperparameter sensitivity test, and the experimental results of the two groups of hyperparameters were close.This shows that the inference method proposed in this paper is insensitive to hyperparameters.Therefore, it is only necessary to select some non-information priors when setting the superparameters of prior distribution.Please refer to page 11 of the revised version.

3. For reproducibility purposes, I would suggest to make the code available (if it is possible) either as supplementary material to be published online or as a reference to a gitHub website.

Response:Thanks for the reviewer’s suggestion! We uploaded all the R codes for the experiment as you suggested.

4. Pages 9-10: It scems unnecessary to replace (θ, σ−2, · · · )with(ζ1, · · · , ζ6).

Response:Thank you for the insightful point. Since the (θ, σ−2, · · · )with superscripts and subscripts

will be confused with the labels in the MCMC algorithm, we replace them with(ζ1, · · · , ζ6),

which will make the MCMC algorithm clearer.

5. For comparison purpose, could you combine Table 1-3 and redesign the table?

Response: We are very grateful to the reviewer for the valuable comment. We deleted Table 2

and redesigned the table to merge Table 3 with Table 1. Please refer to page 9 of the revised

version.

---

## [Decision Letter · Decision Letter 1]

18 Jan 2023

PONE-D-22-18597R1Study of Bayesian variable selection method on mixed linear  regression modelsPLOS ONE

Dear Dr. Li,

Thank you for submitting your manuscript to PLOS ONE. One of reviewer have several suggestions for your manuscript and need a minor revision. Therefore, we invite you to submit a revised version of the manuscript that addresses the points raised during the review process.

We look forward to receiving your revised manuscript.

Kind regards,

Lei Shi

Academic Editor

PLOS ONE

Journal Requirements:

Reviewers' comments:

Reviewer's Responses to Questions

**Comments to the Author**

1. If the authors have adequately addressed your comments raised in a previous round of review and you feel that this manuscript is now acceptable for publication, you may indicate that here to bypass the “Comments to the Author” section, enter your conflict of interest statement in the “Confidential to Editor” section, and submit your "Accept" recommendation.

Reviewer #1: (No Response)

Reviewer #2: All comments have been addressed

2. Is the manuscript technically sound, and do the data support the conclusions?

Reviewer #1: Yes

Reviewer #2: Yes

3. Has the statistical analysis been performed appropriately and rigorously? 

Reviewer #1: N/A

Reviewer #2: Yes

4. Have the authors made all data underlying the findings in their manuscript fully available?

Reviewer #1: Yes

Reviewer #2: Yes

5. Is the manuscript presented in an intelligible fashion and written in standard English?

Reviewer #1: No

Reviewer #2: Yes

6. Review Comments to the Author

Reviewer #1: General comments:

The authors have made a number of changes on the paper and the quality is now much improved. I feel that the paper now is technically acceptable but there are still some problems need to be fixed.

Specific comments:

1. Many mathematical notations are not displayed appropriately, e.g. “expa” should be "\\exp" in Equations (5)—(8); there are typos in Equation (8) and so on. These problems severely affect the readability of the paper. Can you employ LaTex or other professional software to handle mathematical notations?

2. There are some language issues, e.g., “Case analysis” should be “Case study”, “interest covariates” should be “covariates of interest” and so on. Proof read is needed!

3. Please describe how did you select 512 patients from the original data set.

Reviewer #2: (No Response)

7. PLOS authors have the option to publish the peer review history of their article (what does this mean?). If published, this will include your full peer review and any attached files.

Reviewer #1: No

Reviewer #2: No

---

## [Author Response · Author response to Decision Letter 1]

22 Feb 2023

Response to Reviewer 1

1. Many mathematical notations are not displayed appropriately, e.g. “expa” should be “exp” in Equations (5)—(8); there are typos in Equation (8) and so on. These problems severely affect the readability of the paper. Can you employ LaTex or other professional software to handle mathematical notations?

Response: Thank you for your question! We have made modifications according to your suggestions. Please refer to page 5 of the revised version.

2. There are some language issues, e.g., “Case analysis” should be “Case study”, “interest co-variates” should be “covariates of interest” and so on. Proof read is needed!

Response:We are very grateful to the reviewer for the valuable comments. We have carefully proofread and revised the article. We have revised many places and marked them in red in the article.

3. Please describe how did you select 512 patients from the original data set.

Response: Thank you for your valuable question. We have eliminated the individuals with missing information, and the remaining 512 individuals. Please refer to page 12 of the revised version.

---

## [Editor Report · Decision Letter 2]

2 Mar 2023

Study of Bayesian variable selection method on mixed linear  regression models

PONE-D-22-18597R2

Dear Dr. Li,

We’re pleased to inform you that your manuscript has been judged scientifically suitable for publication and will be formally accepted for publication once it meets all outstanding technical requirements.

Kind regards,

Lei Shi

Academic Editor

PLOS ONE
---

## [Editor Report · Acceptance letter]

7 Mar 2023

PONE-D-22-18597R2 

Study of Bayesian variable selection method on mixed linear regression models 

Dear Dr. Li:

I'm pleased to inform you that your manuscript has been deemed suitable for publication in PLOS ONE. Congratulations! Your manuscript is now with our production department. 

Kind regards, 

on behalf of

Dr. Lei Shi 

Academic Editor

PLOS ONE